# Safe and Nested Subgame Solving for Imperfect-Information Games

**Noam Brown**
Computer Science Department
Carnegie Mellon University
Pittsburgh, PA 15217
noamb@cs.cmu.edu

**Tuomas Sandholm**
Computer Science Department
Carnegie Mellon University
Pittsburgh, PA 15217
sandholm@cs.cmu.edu

## Abstract

In imperfect-information games, the optimal strategy in a subgame may depend on the strategy in other, unreached subgames. Thus a subgame cannot be solved in isolation and must instead consider the strategy for the entire game as a whole, unlike perfect-information games. Nevertheless, it is possible to first approximate a solution for the whole game and then improve it in individual subgames. This is referred to as *subgame solving*. We introduce subgame-solving techniques that outperform prior methods both in theory and practice. We also show how to adapt them, and past subgame-solving techniques, to respond to opponent actions that are outside the original action abstraction; this significantly outperforms the prior state-of-the-art approach, action translation. Finally, we show that subgame solving can be repeated as the game progresses down the game tree, leading to far lower exploitability. These techniques were a key component of *Libratus*, the first AI to defeat top humans in heads-up no-limit Texas hold'em poker.

## 1 Introduction

Imperfect-information games model strategic settings that have hidden information. They have a myriad of applications including negotiation, auctions, cybersecurity, and physical security.

In perfect-information games, determining the optimal strategy at a decision point only requires knowledge of the game tree's current node and the remaining game tree beyond that node (the *subgame* rooted at that node). This fact has been leveraged by nearly every AI for perfect-information games, including AIs that defeated top humans in chess [7] and Go [29]. In checkers, the ability to decompose the game into smaller independent subgames was even used to solve the entire game [27]. However, it is not possible to determine a subgame's optimal strategy in an imperfect-information game using only knowledge of that subgame, because the game tree's exact node is typically unknown. Instead, the optimal strategy may depend on the value an opponent could have received in some other, unreached subgame. Although this is counter-intuitive, we provide a demonstration in Section 2.

Rather than rely on subgame decomposition, past approaches for imperfect-information games typically solved the game as a whole upfront. For example, heads-up limit Texas hold'em, a relatively simple form of poker with $10^{13}$ decision points, was essentially solved without decomposition [2]. However, this approach cannot extend to larger games, such as heads-up no-limit Texas hold'em—the primary benchmark in imperfect-information game solving—which has $10^{161}$ decision points [16].

The standard approach to computing strategies in such large games is to first generate an *abstraction* of the game, which is a smaller version of the game that retains as much as possible the strategic characteristics of the original game [24, 26, 25]. For example, a continuous action space might be discretized. This abstract game is solved and its solution is used when playing the full game by mapping states in the full game to states in the abstract game. We refer to the solution of an abstraction (or more generally any approximate solution to a game) as a *blueprint* strategy.

In heavily abstracted games, a blueprint strategy may be far from the true solution. *Subgame solving* attempts to improve upon the blueprint strategy by solving in real time a more fine-grained abstraction for an encountered subgame, while fitting its solution within the overarching blueprint strategy.

## 2 Coin Toss

In this section we provide intuition for why an imperfect-information subgame cannot be solved in isolation. We demonstrate this in a simple game we call Coin Toss, shown in Figure 1a, which will be used as a running example throughout the paper.

Coin Toss is played between players $P_1$ and $P_2$. The figure shows rewards only for $P_1$; $P_2$ always receives the negation of $P_1$'s reward. A coin is flipped and lands either Heads or Tails with equal probability, but only $P_1$ sees the outcome. $P_1$ then chooses between actions "Sell" and "Play." The Sell action leads to a subgame whose details are not important, but the *expected value* (EV) of choosing the Sell action will be important. (For simplicity, one can equivalently assume *in this section* that Sell leads to an immediate terminal reward, where the value depends on whether the coin landed Heads or Tails). If the coin lands Heads, it is considered lucky and $P_1$ receives an EV of $0.50 for choosing Sell. On the other hand, if the coin lands Tails, it is considered unlucky and $P_1$ receives an EV of $-$0.50 for action Sell. (That is, $P_1$ must on average pay $0.50 to get rid of the coin). If $P_1$ instead chooses Play, then $P_2$ may guess how the coin landed. If $P_2$ guesses correctly, then $P_1$ receives a reward of $-$1. If $P_2$ guesses incorrectly, then $P_1$ receives $1. $P_2$ may also forfeit, which should never be chosen but will be relevant in later sections. We wish to determine the optimal strategy for $P_2$ in the subgame $S$ that occurs after $P_1$ chooses Play, shown in Figure 1a.

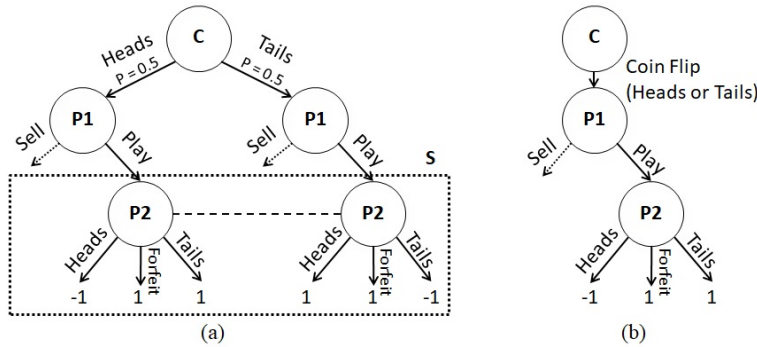

Figure 1: (a) The example game of Coin Toss. "C" represents a chance node. $S$ is a Player 2 ($P_2$) subgame. The dotted line between the two $P_2$ nodes means that $P_2$ cannot distinguish between them. (b) The public game tree of Coin Toss. The two outcomes of the coin flip are only observed by $P_1$.

Were $P_2$ to always guess Heads, $P_1$ would receive $0.50 for choosing Sell when the coin lands Heads, and $1 for Play when it lands Tails. This would result in an average of $0.75 for $P_1$. Alternatively, were $P_2$ to always guess Tails, $P_1$ would receive $1 for choosing Play when the coin lands Heads, and $-$0.50 for choosing Sell when it lands Tails. This would result in an average reward of $0.25 for $P_1$. However, $P_2$ would do even better by guessing Heads with 25% probability and Tails with 75% probability. In that case, $P_1$ could only receive $0.50 (on average) by choosing Play when the coin lands Heads—the same value received for choosing Sell. Similarly, $P_1$ could only receive $-$0.50 by choosing Play when the coin lands Tails, which is the same value received for choosing Sell. This would yield an average reward of $0 for $P_1$. It is easy to see that this is the best $P_2$ can do, because $P_1$ can average $0 by always choosing Sell. Therefore, choosing Heads with 25% probability and Tails with 75% probability is an optimal strategy for $P_2$ in the "Play" subgame.

Now suppose the coin is considered lucky if it lands Tails and unlucky if it lands Heads. That is, the expected reward for selling the coin when it lands Heads is now $-$0.50 and when it lands Tails is now $0.50. It is easy to see that $P_2$'s optimal strategy for the "Play" subgame is now to guess Heads with 75% probability and Tails with 25% probability. This shows that a player's optimal strategy in a subgame can depend on the strategies and outcomes in other parts of the game. Thus, one cannot solve a subgame using information about that subgame alone. This is the central challenge of imperfect-information games as opposed to perfect-information games.

# 3 Notation and Background

In a two-player zero-sum extensive-form game there are two players, $\mathcal{P} = \{1, 2\}$. $H$ is the set of all possible nodes, represented as a sequence of actions. $A(h)$ is the actions available in a node and $P(h) \in \mathcal{P} \cup c$ is the player who acts at that node, where $c$ denotes chance. Chance plays an action $a \in A(h)$ with a fixed probability. If action $a \in A(h)$ leads from $h$ to $h'$, then we write $h \cdot a = h'$. If a sequence of actions leads from $h$ to $h'$, then we write $h \sqsubset h'$. The set of nodes $Z \subseteq H$ are terminal nodes. For each player $i \in \mathcal{P}$, there is a payoff function $u_i : Z \to \Re$ where $u_1 = -u_2$.

Imperfect information is represented by *information sets* (infosets). Every node $h \in H$ belongs to exactly one infoset for each player. For any infoset $I_i$, nodes $h, h' \in I_i$ are indistinguishable to player $i$. Thus the same player must act at all the nodes in an infoset, and the same actions must be available. Let $P(I_i)$ and $A(I_i)$ be such that all $h \in I_i$, $P(I_i) = P(h)$ and $A(I_i) = A(h)$.

A strategy $\sigma_i(I_i)$ is a probability vector over $A(I_i)$ for infosets where $P(I_i) = i$. The probability of action $a$ is denoted by $\sigma_i(I_i, a)$. For all $h \in I_i$, $\sigma_i(h) = \sigma_i(I_i)$. A full-game strategy $\sigma_i \in \Sigma_i$ defines a strategy for each player $i$ infoset. A strategy profile $\sigma$ is a tuple of strategies, one for each player. The expected payoff for player $i$ if all players play the strategy profile $\langle \sigma_i, \sigma_{-i} \rangle$ is $u_i(\sigma_i, \sigma_{-i})$, where $\sigma_{-i}$ denotes the strategies in $\sigma$ of all players other than $i$.

Let $\pi^\sigma(h) = \prod_{h' \cdot a \sqsubseteq h} \sigma_{P(h')}(h', a)$ denote the probability of reaching $h$ if all players play according to $\sigma$. $\pi_i^\sigma(h)$ is the contribution of player $i$ to this probability (that is, the probability of reaching $h$ if chance and all players other than $i$ always chose actions leading to $h$). $\pi_{-i}^\sigma(h)$ is the contribution of all players, and chance, *other than* $i$. $\pi^\sigma(h, h')$ is the probability of reaching $h'$ given that $h$ has been reached, and 0 if $h \not\sqsubset h'$. This papers focuses on *perfect-recall* games, where a player never forgets past information. Thus, for every $I_i$, $\forall h, h' \in I_i$, $\pi_i^\sigma(h) = \pi_i^\sigma(h')$. We define $\pi_i^\sigma(I_i) = \pi_i^\sigma(h)$ for $h \in I_i$. Also, $I_i' \sqsubset I_i$ if for some $h' \in I_i'$ and some $h \in I_i$, $h' \sqsubset h$. Similarly, $I_i' \cdot a \sqsubset I_i$ if $h' \cdot a \sqsubset h$.

A *Nash equilibrium* [22] is a strategy profile $\sigma^*$ where no player can improve by shifting to a different strategy, so $\sigma^*$ satisfies $\forall i$, $u_i(\sigma_i^*, \sigma_{-i}^*) = \max_{\sigma_i' \in \Sigma_i} u_i(\sigma_i', \sigma_{-i}^*)$. A *best response* $BR(\sigma_{-i})$ is a strategy for player $i$ that is optimal against $\sigma_{-i}$. Formally, $BR(\sigma_{-i})$ satisfies $u_i(BR(\sigma_{-i}), \sigma_{-i}) = \max_{\sigma_i' \in \Sigma_i} u_i(\sigma_i', \sigma_{-i})$. In a two-player zero-sum game, the *exploitability* $exp(\sigma_i)$ of a strategy $\sigma_i$ is how much worse $\sigma_i$ does against an opponent best response than a Nash equilibrium strategy would do. Formally, exploitability of $\sigma_i$ is $u_i(\sigma^*) - u_i(\sigma_i, BR(\sigma_i))$, where $\sigma^*$ is a Nash equilibrium.

The expected *value* of a node $h$ when players play according to $\sigma$ is $v_i^\sigma(h) = \sum_{z \in Z} \left( \pi^\sigma(h, z) u_i(z) \right)$. An infoset's value is the weighted average of the values of the nodes in the infoset, where a node is weighed by the player's belief that she is in that node. Formally, $v_i^\sigma(I_i) = \frac{\sum_{h \in I_i} \left( \pi_{-i}^\sigma(h) v_i^\sigma(h) \right)}{\sum_{h \in I_i} \pi_{-i}^\sigma(h)}$ and $v_i^\sigma(I_i, a) = \frac{\sum_{h \in I_i} \left( \pi_{-i}^\sigma(h) v_i^\sigma(h \cdot a) \right)}{\sum_{h \in I_i} \pi_{-i}^\sigma(h)}$. A *counterfactual best response* [21] $CBR(\sigma_{-i})$ is a best response that also maximizes value in unreached infosets. Specifically, a counterfactual best response is a best response $\sigma_i$ with the additional condition that if $\sigma_i(I_i, a) > 0$ then $v_i^\sigma(I_i, a) = \max_{a'} v_i^\sigma(I_i, a')$. We further define *counterfactual best response value* $CBV^{\sigma_{-i}}(I_i)$ as the value player $i$ expects to achieve by playing according to $CBR(\sigma_{-i})$, having already reached infoset $I_i$. Formally, $CBV^{\sigma_{-i}}(I_i) = v_i^{\langle CBR(\sigma_{-i}), \sigma_{-i} \rangle}(I_i)$ and $CBV^{\sigma_{-i}}(I_i, a) = v_i^{\langle CBR(\sigma_{-i}), \sigma_{-i} \rangle}(I_i, a)$.

An *imperfect-information subgame*, which we refer to simply as a *subgame* in this paper, can in most cases (but not all) be described as including all nodes which share prior *public* actions (that is, actions viewable to both players). In poker, for example, a subgame is uniquely defined by a sequence of bets and public board cards. Figure 1b shows the public game tree of Coin Toss. Formally, an imperfect-information subgame is a set of nodes $S \subseteq H$ such that for all $h \in S$, if $h \sqsubset h'$, then $h' \in S$, and for all $h \in S$ and all $i \in \mathcal{P}$, if $h' \in I_i(h)$ then $h' \in S$. Define $S_{top}$ as the set of earliest-reachable nodes in $S$. That is, $h \in S_{top}$ if $h \in S$ and $h' \notin S$ for any $h' \sqsubset h$.

# 4 Prior Approaches to Subgame Solving

This section reviews prior techniques for subgame solving in imperfect-information games, which we build upon. Throughout this section, we refer to the Coin Toss game shown in Figure 1a.

As discussed in Section 1, a standard approach to dealing with large imperfect-information games is to solve an abstraction of the game. The abstract solution is a (probably suboptimal) strategy profile

in the full game. We refer to this full-game strategy profile as the blueprint. The goal of subgame solving is to improve upon the blueprint by changing the strategy only in a subgame.

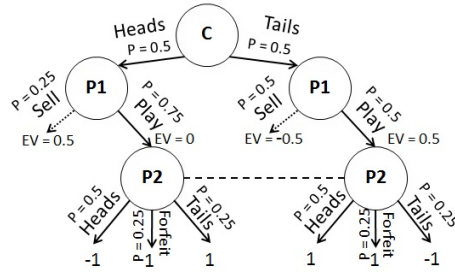

Figure 2: The blueprint strategy we refer to in the game of Coin Toss. The Sell action leads to a subgame that is not displayed. Probabilities are shown for all actions. The dotted line means the two $P_2$ nodes share an infoset. The EV of each $P_1$ action is also shown.

Assume that a blueprint strategy profile $\sigma$ (shown in Figure 2) has already been computed for Coin Toss in which $P_1$ chooses Play $\frac{3}{4}$ of the time with Heads and $\frac{1}{2}$ of the time with Tails, and $P_2$ chooses Heads $\frac{1}{2}$ of the time, Tails $\frac{1}{4}$ of the time, and Forfeit $\frac{1}{4}$ of the time after $P_1$ chooses Play. The details of the blueprint strategy in the Sell subgame are not relevant in this section, but the EV for choosing the Sell action *is* relevant. We assume that if $P_1$ chose the Sell action and played optimally thereafter, then she would receive an expected payoff of $0.5$ if the coin is Heads, and $-0.5$ if the coin is Tails. We will attempt to improve $P_2$'s strategy in the subgame $S$ that follows $P_1$ choosing Play.

## 4.1 Unsafe Subgame Solving

We first review the most intuitive form of subgame solving, which we refer to as *Unsafe subgame solving* [1, 12, 13, 10]. This form of subgame solving assumes both players played according to the blueprint strategy prior to reaching the subgame. That defines a probability distribution over the nodes at the root of the subgame $S$, representing the probability that the true game state matches that node. A strategy for the subgame is then calculated which assumes that this distribution is correct.

In all subgame solving algorithms, an *augmented subgame* containing $S$ and a few additional nodes is solved to determine the strategy for $S$. Applying Unsafe subgame solving to the blueprint strategy in Coin Toss (after $P_1$ chooses Play) means solving the augmented subgame shown in Figure 3a.

Specifically, the augmented subgame consists of only an initial chance node and $S$. The initial chance node reaches $h \in S_{top}$ with probability $\frac{\pi^\sigma(h)}{\sum_{h' \in S_{top}} \pi^\sigma(h')}$. The augmented subgame is solved and its strategy for $P_2$ is used in $S$ rather than the blueprint strategy.

Unsafe subgame solving lacks theoretical solution quality guarantees and there are many situations where it performs extremely poorly. Indeed, if it were applied to the blueprint strategy of Coin Toss then $P_2$ would always choose Heads—which $P_1$ could exploit severely by only choosing Play with Tails. Despite the lack of theoretical guarantees and potentially bad performance, Unsafe subgame solving is simple and can *sometimes* produce low-exploitability strategies, as we show later.

We now move to discussing *safe* subgame-solving techniques, that is, ones that ensure that the exploitability of the strategy is no higher than that of the blueprint strategy.

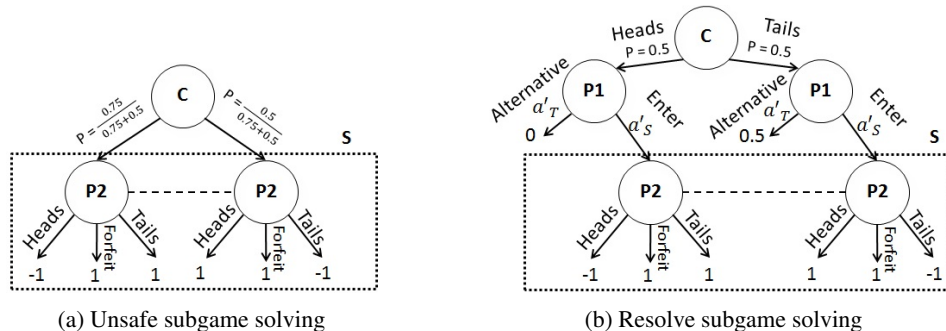

(a) Unsafe subgame solving      (b) Resolve subgame solving

Figure 3: The augmented subgames solved to find a $P_2$ strategy in the Play subgame of Coin Toss.

## 4.2 Subgame Resolving

In *subgame Resolving* [6], a safe strategy is computed for $P_2$ in the subgame by solving the augmented subgame shown in Figure 3b, producing an equilibrium strategy $\sigma^S$. This augmented subgame differs from Unsafe subgame solving by giving $P_1$ the option to "opt out" from entering $S$ and instead receive the EV of playing optimally against $P_2$'s blueprint strategy in $S$.

Specifically, the augmented subgame for Resolving differs from unsafe subgame solving as follows. For each $h_{top} \in S_{top}$ we insert a new $P_1$ node $h_r$, which exists only in the augmented subgame, between the initial chance node and $h_{top}$. The set of these $h_r$ nodes is $S_r$. The initial chance node connects to each node $h_r \in S_r$ in proportion to the probability that player $P_1$ could reach $h_{top}$ if $P_1$ tried to do so (that is, in proportion to $\pi^\sigma_{-1}(h_{top})$). At each node $h_r \in S_r$, $P_1$ has two possible actions. Action $a'_S$ leads to $h_{top}$, while action $a'_T$ leads to a terminal payoff that awards the value of playing optimally against $P_2$'s blueprint strategy, which is $CBV^{\sigma_2}(I_1(h_{top}))$. In the blueprint strategy of Coin Toss, $P_1$ choosing Play after the coin lands Heads results in an EV of 0, and $\frac{1}{2}$ if the coin is Tails. Therefore, $a'_T$ leads to a terminal payoff of 0 for Heads and $\frac{1}{2}$ for Tails. After the equilibrium strategy $\sigma^S$ is computed in the augmented subgame, $P_2$ plays according to the computed subgame strategy $\sigma^S_2$ rather than the blueprint strategy when in $S$. The $P_1$ strategy $\sigma^S_1$ is not used.

Clearly $P_1$ cannot do worse than always picking action $a'_T$ (which awards the highest EV $P_1$ could achieve against $P_2$'s blueprint). But $P_1$ also cannot do *better* than always picking $a'_T$, because $P_2$ could simply play according to the blueprint in $S$, which means action $a'_S$ would give the same EV to $P_1$ as action $a'_T$ (if $P_1$ played optimally in $S$). In this way, the strategy for $P_2$ in $S$ is pressured to be no worse than that of the blueprint. In Coin Toss, if $P_2$ were to always choose Heads (as was the case in Unsafe subgame solving), then $P_1$ would always choose $a'_T$ with Heads and $a'_S$ with Tails.

Resolving guarantees that $P_2$'s exploitability will be no higher than the blueprint's (and may be better). However, it may miss opportunities for improvement. For example, if we apply Resolving to the example blueprint in Coin Toss, one solution to the augmented subgame is the blueprint itself, so $P_2$ may choose Forfeit $25\%$ of the time even though Heads and Tails dominate that action. Indeed, the original purpose of Resolving was not to *improve* upon a blueprint strategy in a subgame, but rather to compactly store it by keeping only the EV at the root of the subgame and then reconstructing the strategy in real time when needed rather than storing the whole subgame strategy.

Maxmargin subgame solving [21], discussed in Appendix A, can improve performance by defining a *margin* $M^{\sigma^S}(I_1) = CBV^{\sigma_2}(I_1) - CBV^{\sigma^S_2}(I_1)$ for each $I_1 \in S_{top}$ and maximizing $\min_{I_1 \in S_{top}} M^{\sigma^S}(I_1)$. Resolving only makes all margins nonnegative. However, Maxmargin does worse in practice when using estimates of equilibrium values as discussed in Appendix C.

## 5 Reach Subgame Solving

All of the subgame-solving techniques described in Section 4 only consider the target subgame in isolation, which can lead to suboptimal strategies. For example, Maxmargin solving applied to $S$ in Coin Toss results in $P_2$ choosing Heads with probability $\frac{5}{8}$ and Tails with $\frac{3}{8}$ in $S$. This results in $P_1$ receiving an EV of $-\frac{1}{4}$ by choosing Play in the Heads state, and an EV of $\frac{1}{4}$ in the Tails state. However, $P_1$ could simply always choose Sell in the Heads state (earning an EV of $0.5$) and Play in the Tails state and receive an EV of $\frac{3}{8}$ for the entire game. In this section we introduce *Reach subgame solving*, an improvement to past subgame-solving techniques that considers *what the opponent could have alternatively received from other subgames*.[1] For example, a better strategy for $P_2$ would be to choose Heads with probability $\frac{3}{4}$ and Tails with probability $\frac{1}{4}$. Then $P_1$ is indifferent between choosing Sell and Play in both cases and overall receives an expected payoff of 0 for the whole game.

However, that strategy is only optimal if $P_1$ would indeed achieve an EV of $0.5$ for choosing Sell in the Heads state and $-0.5$ in the Tails state. That would be the case if $P_2$ played according to the blueprint in the Sell subgame (which is not shown), but in reality we would apply subgame solving to the Sell subgame if the Sell action were taken, which would change $P_2$'s strategy there and therefore $P_1$'s EVs. Applying subgame solving to any subgame encountered during play is equivalent to applying it to all subgames independently; ultimately, the same strategy is played in both cases. Thus, we must consider that the EVs from other subgames may differ from what the blueprint says because subgame solving would be applied to them as well.

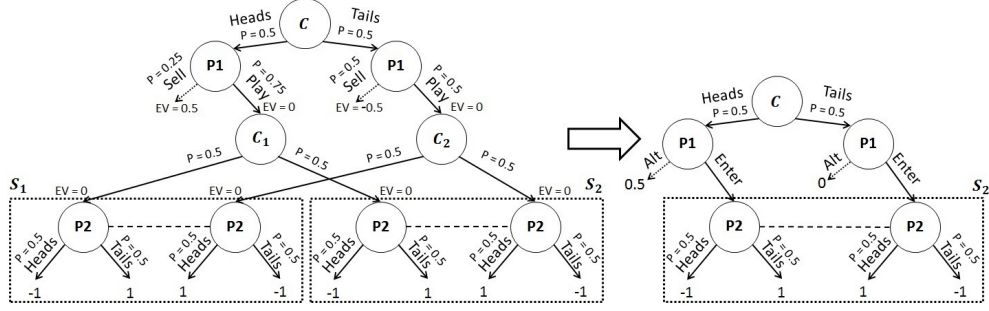

Figure 4: Left: A modified game of Coin Toss with two subgames. The nodes $C_1$ and $C_2$ are public chance nodes whose outcomes are seen by both $P_1$ and $P_2$. Right: An augmented subgame for one of the subgames according to Reach subgame solving. If only one of the subgames is being solved, then the alternative payoff for Heads can be at most 1. However, if both are solved independently, then the gift must be split among the subgames and must sum to at most 1. For example, the alternative payoff in both subgames can be 0.5.

As an example of this issue, consider the game shown in Figure 4 which contains two identical subgames $S_1$ and $S_2$ where the blueprint has $P_2$ pick Heads and Tails with 50% probability. The Sell action leads to an EV of 0.5 from the Heads state, while Play leads to an EV of 0. If we were to solve just $S_1$, then $P_2$ could afford to always choose Tails in $S_1$, thereby letting $P_1$ achieve an EV of 1 for reaching that subgame from Heads because, due to the chance node $C_1$, $S_1$ is only reached with 50% probability. Thus, $P_1$'s EV for choosing Play would be 0.5 from Heads and $-0.5$ from Tails, which is optimal. We can achieve this strategy in $S_1$ by solving an augmented subgame in which the alternative payoff for Heads is 1. In that augmented subgame, $P_2$ always choosing Tails would be a solution (though not the only solution).

However, if the same reasoning were applied independently to $S_2$ as well, then $P_2$ might always choose Tails in both subgames and $P_1$'s EV for choosing Play from Heads would become 1 while the EV for Sell would only be 0.5. Instead, we could allow $P_1$ to achieve an EV of 0.5 for reaching each subgame from Heads (by setting the alternative payoff for Heads to 0.5). In that case, $P_1$'s overall EV for choosing Play could only increase to 0.5, even if both $S_1$ and $S_2$ were solved independently.

We capture this intuition by considering for each $I_1 \in S_{top}$ all the infosets and actions $I_1' \cdot a' \sqsubset I_1$ that $P_1$ would have taken along the path to $I_1$. If, at some $I_1' \cdot a' \sqsubset I_1$ where $P_1$ acted, there was a different action $a^* \in A(I_1')$ that leads to a higher EV, then $P_1$ would have taken a suboptimal action if they reached $I_1$. The difference in value between $a^*$ and $a'$ is referred to as a *gift*. We can afford to let $P_1$'s value for $I_1$ increase beyond the blueprint value (and in the process lower $P_1$'s value in some other infoset in $S_{top}$), so long as the increase to $I_1$'s value is small enough that choosing actions leading to $I_1$ is still suboptimal for $P_1$. Critically, we must ensure that the increase in value is small enough even when the potential increase across all subgames is summed together, as in Figure 4.[2]

A complicating factor is that gifts we assumed were present may actually not exist. For example, in Coin Toss, suppose applying subgame solving to the Sell subgame results in $P_1$'s value for Sell from the Heads state decreasing from 0.5 to 0.25. If we independently solve the Play subgame, we have no way of knowing that $P_1$'s value for Sell is lower than the blueprint suggested, so we may still assume there is a gift of 0.5 from the Heads state based on the blueprint. Thus, in order to guarantee a theoretical result on exploitability that is as strong as possible, we use in our theory and experiments a *lower bound* on what gifts could be after subgame solving was applied to all other subgames.

Formally, let $\sigma_2$ be a $P_2$ blueprint and let $\sigma_2^{-S}$ be the $P_2$ strategy that results from applying subgame solving independently to a set of disjoint subgames other than $S$. Since we do not want to compute $\sigma_2^{-S}$ in order to apply subgame solving to $S$, let $\lfloor g^{\sigma_2^{-S}}(I_1', a') \rfloor$ be a lower bound of $CBV^{\sigma_2^{-S}}(I_1') - CBV^{\sigma_2^{-S}}(I_1', a')$ that does not require knowledge of $\sigma_2^{-S}$. In our experiments we

use $\lfloor g^{\sigma_2^{-S}}(I_1', a') \rfloor = \max_{a \in A_z(I_1') \cup \{a'\}} CBV^{\sigma_2}(I_1', a) - CBV^{\sigma_2}(I_1', a')$ where $A_z(I_1') \subseteq A(I_1')$ is the set of actions leading immediately to terminal nodes. Reach subgame solving modifies the augmented subgame in Resolving and Maxmargin by increasing the alternative payoff for infoset $I_1 \in S_{top}$ by $\sum_{I_1' \cdot a' \sqsubseteq I_1 | P(I_1') = P_1} \lfloor g^{\sigma_2^{-S}}(I_1', a') \rfloor$. Formally, we define a *reach margin* as

$$M_r^{\sigma^S}(I_1) = M^{\sigma^S}(I_1) + \sum_{I_1' \cdot a' \sqsubseteq I_1 | P(I_1') = P_1} \lfloor g^{\sigma_2^{-S}}(I_1', a') \rfloor \tag{1}$$

This margin is larger than or equal to the one for Maxmargin, because $\lfloor g^{\sigma_2^{-S}}(I', a') \rfloor$ is nonnegative. We refer to the modified algorithms as Reach-Resolve and Reach-Maxmargin.

Using a lower bound on gifts is not necessary to guarantee safety. So long as we use a gift value $g^{\sigma'}(I_1', a') \leq CBV^{\sigma_2}(I_1') - CBV^{\sigma_2}(I_1', a')$, the resulting strategy will be safe. However, using a lower bound further guarantees a reduction to exploitability when a $P_1$ best response reaches with positive probability an infoset $I_1 \in S_{top}$ that has positive margin, as proven in Theorem 1. In practice, it may be best to use an accurate estimate of gifts. One option is to use $\hat{g}^{\sigma_2^{-S}}(I_1', a') = C\tilde{B}V^{\sigma_2}(I_1') - C\tilde{B}V^{\sigma_2}(I_1', a')$ in place of $\lfloor g^{\sigma_2^{-S}}(I_1', a') \rfloor$, where $C\tilde{B}V^{\sigma_2}$ is the closest $P_1$ can get to the value of a counterfactual best response while $P_1$ is constrained to playing within the abstraction that generated the blueprint. Using estimates is covered in more detail in Appendix C.

Theorem 1 shows that when subgames are solved independently and using lower bounds on gifts, Reach-Maxmargin solving has exploitability lower than or equal to past safe techniques. The theorem statement is similar to that of Maxmargin [21], but the margins are now larger (or equal) in size.

**Theorem 1.** *Given a strategy $\sigma_2$ in a two-player zero-sum game, a set of disjoint subgames $\mathbb{S}$, and a strategy $\sigma_2^S$ for each subgame $S \in \mathbb{S}$ produced via Reach-Maxmargin solving using lower bounds for gifts, let $\sigma_2'$ be the strategy that plays according to $\sigma_2^S$ for each subgame $S \in \mathbb{S}$, and $\sigma_2$ elsewhere. Moreover, let $\sigma_2^{-S}$ be the strategy that plays according to $\sigma_2'$ everywhere except for $P_2$ nodes in $S$, where it instead plays according to $\sigma_2$. If $\pi_1^{BR(\sigma_2')}(I_1) > 0$ for some $I_1 \in S_{top}$, then $exp(\sigma_2') \leq exp(\sigma_2^{-S}) - \sum_{h \in I_1} \pi_{-1}^{\sigma_2}(h) M_r^{\sigma^S}(I_1)$.*

So far the described techniques have guaranteed a reduction in exploitability over the blueprint by setting the value of $a_T'$ equal to the value of $P_1$ playing optimally to $P_2$'s blueprint. Relaxing this guarantee by instead setting the value of $a_T'$ equal to an *estimate* of $P_1$'s value when *both* players play optimally leads to far lower exploitability in practice. We discuss this approach in Appendix C.

## 6 Nested Subgame Solving

As we have discussed, large games must be abstracted to reduce the game to a tractable size. This is particularly common in games with large or continuous action spaces. Typically the action space is discretized by action abstraction so that only a few actions are included in the abstraction. While we might limit ourselves to the actions we included in the abstraction, an opponent might choose actions that are not in the abstraction. In that case, the *off-tree* action can be mapped to an action that is in the abstraction, and the strategy from that in-abstraction action can be used. For example, in an auction game we might include a bid of \$100 in our abstraction. If a player bids \$101, we simply treat that as a bid of \$100. This is referred to as *action translation* [14, 28, 8]. Action translation is the state-of-the-art prior approach to dealing with this issue. It has been used, for example, by all the leading competitors in the Annual Computer Poker Competition (ACPC).

In this section, we develop techniques for applying subgame solving to calculate responses to opponent off-tree actions, thereby obviating the need for action translation. That is, rather than simply treat a bid of \$101 as \$100, we calculate in real time a unique response to the bid of \$101. This can also be done in a nested fashion in response to subsequent opponent off-tree actions. Additionally, these techniques can be used to solve finer-grained models as play progresses down the game tree.

We refer to the first method as the *inexpensive* method.[3] When $P_1$ chooses an off-tree action $a$, a subgame $S$ is generated following that action such that for any infoset $I_1$ that $P_1$ might be in, $I_1 \cdot a \in S_{top}$. This subgame may itself be an abstraction. A solution $\sigma^S$ is computed via subgame solving, and $\sigma^S$ is combined with $\sigma$ to form a new blueprint $\sigma'$ in the expanded abstraction that now includes action $a$. The process repeats whenever $P_1$ again chooses an off-tree action.

To conduct safe subgame solving in response to off-tree action $a$, we could calculate $CBV^{\sigma_2}(I_1, a)$ by defining, via action translation, a $P_2$ blueprint following $a$ and best responding to it [4]. However, that could be computationally expensive and would likely perform poorly in practice because, as we show later, action translation is highly exploitable. Instead, we relax the guarantee of safety and use $C\tilde{B}V^{\sigma_2}(I_1)$ for the alternative payoff, where $C\tilde{B}V^{\sigma_2}(I_1)$ is $P_1$'s counterfactual best response value in $I_1$ when constrained to playing in the blueprint abstraction (which excludes action $a$). In this case, exploitability depends on how well $C\tilde{B}V^{\sigma_2}(I_1)$ approximates $CBV^{\sigma_2^*}(I_1)$, where $\sigma_2^*$ is an optimal $P_2$ strategy (see Appendix C).[4] In general, we find that only a small number of near-optimal actions need to be included in the blueprint abstraction for $C\tilde{B}V^{\sigma_2}(I_1)$ to be close to $CBV^{\sigma_2^*}(I_1)$. We can then approximate a near-optimal response to any opponent action, even in a continuous action space.

The "inexpensive" approach cannot be combined with Unsafe subgame solving because the probability of reaching an action outside of a player's abstraction is undefined. Nevertheless, a similar approach is possible with Unsafe subgame solving (as well as all the other subgame-solving techniques) by starting the subgame solving at $h$ rather than at $h \cdot a$. In other words, if action $a$ taken in node $h$ is not in the abstraction, then Unsafe subgame solving is conducted in the smallest subgame containing $h$ (and action $a$ is added to that abstraction). This increases the size of the subgame compared to the inexpensive method because a strategy must be recomputed for every action $a' \in A(h)$ in addition to $a$. We therefore call this method the *expensive* method. We present experiments with both methods.

## 7 Experiments

Our experiments were conducted on heads-up no-limit Texas hold'em, as well as two smaller-scale poker games we call *No-Limit Flop Hold'em* (NLFH) and *No-Limit Turn Hold'em* (NLTH). The description for these games can be found in Appendix G. For equilibrium finding, we used CFR+ [30].

Our first experiment compares the performance of the subgame-solving techniques when applied to information abstraction (which is card abstraction in the case of poker). Specifically, we solve NLFH with no information abstraction on the preflop. On the flop, there are 1,286,792 infosets for each betting sequence; the abstraction buckets them into 200, 2,000, or 30,000 abstract ones (using a leading information abstraction algorithm [9]). We then apply subgame solving immediately after the flop community cards are dealt. We experiment with two versions of the game, one small and one large, which include only a few of the available actions in each infoset. We also experimented on abstractions of NLTH. In that case, we solve NLTH with no information abstraction on the preflop or flop. On the turn, there are 55,190,538 infosets for each betting sequence; the abstraction buckets them into 200, 2,000, or 20,000 abstract ones. We apply subgame solving immediately after the turn community card is dealt. Table 1 shows the performance of each technique when using 30,000 buckets (20,000 for NLTH). The full results are presented in Appendix E. In all our experiments, exploitability is measured in the standard units used in this field: milli big blinds per hand (mbb/h).

|  | Small Flop Holdem | Large Flop Holdem | Turn Holdem |
|---|---|---|---|
| Blueprint Strategy | 91.28 | 41.41 | 345.5 |
| Unsafe | 5.514 | 396.8 | 79.34 |
| Resolve | 54.07 | 23.11 | 251.8 |
| Maxmargin | 43.43 | 19.50 | 234.4 |
| Reach-Maxmargin | 41.47 | 18.80 | 233.5 |
| Reach-Maxmargin (no split) | 25.88 | 16.41 | 175.5 |
| Estimate | 24.23 | 30.09 | 76.44 |
| Estimate+Distributional | 34.30 | 10.54 | 74.35 |
| Reach-Estimate+Distributional | 22.58 | 9.840 | 72.59 |
| Reach-Estimate+Distributional (no split) | 17.33 | 8.777 | 70.68 |

Table 1: Exploitability of various subgame-solving techniques in three different games.

Estimate and Estimate+Distributional are techniques introduced in Appendix C. We use a normal distribution in the Distributional subgame solving experiments, with standard deviation determined by the heuristic presented in Appendix C.1.

Since subgame solving begins immediately after a chance node with an extremely high branching factor ($1,755$ in NLFH), the gifts for the Reach algorithms are divided among subgames inefficiently.

Many subgames do not use the gifts at all, while others could make use of more. In the experiments we show results both for the theoretically safe splitting of gifts, as well as a more aggressive version where gifts are scaled up by the branching factor of the chance node $(1,755)$. This weakens the theoretical guarantees of the algorithm, but in general did better than splitting gifts in a theoretically correct manner. However, this is not universally true. Appendix F shows that in at least one case, exploitability increased when gifts were scaled up too aggressively. In all cases, using Reach subgame solving in at least the theoretical safe method led to lower exploitability.

Despite lacking theoretical guarantees, Unsafe subgame solving did surprisingly well in most games. However, it did substantially worse in Large NLFH with 30,000 buckets. This exemplifies its variability. Among the safe methods, all of the changes we introduce show improvement over past techniques. The Reach-Estimate + Distributional algorithm generally resulted in the lowest exploitability among the various choices, and in most cases beat unsafe subgame solving.

The second experiment evaluates nested subgame solving, and compares it to action translation. In order to also evaluate action translation, in this experiment, we create an NLFH game that includes 3 bet sizes at every point in the game tree (0.5, 0.75, and 1.0 times the size of the pot); a player can also decide not to bet. Only one bet (i.e., no raises) is allowed on the preflop, and three bets are allowed on the flop. There is no information abstraction anywhere in the game. We also created a second, smaller abstraction of the game in which there is still no information abstraction, but the $0.75\times$ pot bet is never available. We calculate the exploitability of one player using the smaller abstraction, while the other player uses the larger abstraction. Whenever the large-abstraction player chooses a $0.75\times$ pot bet, the small-abstraction player generates and solves a subgame for the remainder of the game (which again does not include any subsequent $0.75\times$ pot bets) using the nested subgame-solving techniques described above. This subgame strategy is then used as long as the large-abstraction player plays within the small abstraction, but if she chooses the $0.75\times$ pot bet again later, then the subgame solving is used again, and so on.

Table 2 shows that all the subgame-solving techniques substantially outperform action translation. We did not test distributional alternative payoffs in this experiment, since the calculated best response values are likely quite accurate. These results suggest that nested subgame solving is preferable to action translation (if there is sufficient time to solve the subgame).

|  | mbb/h |
| --- | --- |
| Randomized Pseudo-Harmonic Mapping | 1,465 |
| Resolve | 150.2 |
| Reach-Maxmargin (Expensive) | 149.2 |
| Unsafe (Expensive) | 148.3 |
| Maxmargin | 122.0 |
| Reach-Maxmargin | 119.1 |

Table 2: Exploitability of the various subgame-solving techniques in nested subgame solving. The performance of the pseudo-harmonic action translation is also shown.

We used the techniques presented in this paper to develop *Libratus*, an AI that competed against four top human professionals in heads-up no-limit Texas hold'em [5]. Heads-up no-limit Texas hold'em has been the primary benchmark challenge for AI in imperfect-information games. The competition involved 120,000 hands of poker and a prize pool of $200,000 split among the humans to incentivize strong play. The AI decisively defeated the human team by 147 mbb / hand, with 99.98% statistical significance. This was the first, and so far only, time an AI defeated top humans in no-limit poker.

## 8 Conclusion

We introduced a subgame-solving technique for imperfect-information games that has stronger theoretical guarantees and better practical performance than prior subgame-solving methods. We presented results on exploitability of both safe and unsafe subgame-solving techniques. We also introduced a method for nested subgame solving in response to the opponent's off-tree actions, and demonstrated that this leads to dramatically better performance than the usual approach of action translation. This is, to our knowledge, the first time that exploitability of subgame-solving techniques has been measured in large games.

Finally, we demonstrated the effectiveness of these techniques in practice in heads-up no-limit Texas Hold'em poker, the main benchmark challenge for AI in imperfect-information games. We developed the first AI to reach the milestone of defeating top humans in heads-up no-limit Texas Hold'em.

## 9 Acknowledgments

This material is based on work supported by the National Science Foundation under grants IIS-1718457, IIS-1617590, and CCF-1733556, and the ARO under award W911NF-17-1-0082, as well as XSEDE computing resources provided by the Pittsburgh Supercomputing Center. The *Brains vs. AI* competition was sponsored by Carnegie Mellon University, Rivers Casino, GreatPoint Ventures, Avenue4Analytics, TNG Technology Consulting, Artificial Intelligence, Intel, and Optimized Markets, Inc. We thank Kristen Gardner, Marcelo Gutierrez, Theo Gutman-Solo, Eric Jackson, Christian Kroer, Tim Reiff, and the anonymous reviewers for helpful feedback.

## Footnotes

[1]Other subgame-solving methods have also considered the cost of reaching a subgame [31, 15]. However, those approaches are not correct in theory when applied in real time to any subgame reached during play.

[2]In this paper and in our experiments, we allow any infoset that descends from a gift to increase by the size of the gift (e.g., in Figure 4 the gift from Heads is 0.5, so we allow $P_1$'s value for Heads in both $S_1$ and $S_2$ to increase by 0.5). However, any division of the gift among subgames is acceptable so long as the potential increase across all subgames (multiplied by the probability of $P_1$ reaching that subgame) does not exceed the original gift. For example in Figure 4 if we only apply Reach subgame solving to $S_1$, then we could allow the Heads state in $S_1$ to increase by 1 rather than just by 0.5. In practice, some divisions may do better than others. The division we use in this paper (applying gifts equally to all subgames) did well in practice.

[3]Following our study, the AI DeepStack used a technique similar to this form of nested subgame solving [20].

[4]We estimate $CBV^{\sigma_2^*}(I_1)$ rather than $CBV^{\sigma_2^*}(I_1, a)$ because $CBV^{\sigma_2^*}(I_1) - CBV^{\sigma_2^*}(I_1, a)$ is a gift that may be added to the alternative payoff anyway.

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
