[Supplementary Material]

# Appendix: Supplementary Material

## A  Maxmargin Solving

*Maxmargin solving* [21] is similar to Resolving, except that it seeks to improve $P_2$'s strategy in the subgame strategy as much as possible. While Resolving seeks a strategy for $P_2$ in $S$ that would simply dissuade $P_1$ from entering $S$, Maxmargin solving additionally seeks to punish $P_1$ as much as possible if $P_1$ nevertheless chooses to enter $S$. A *subgame margin* is defined for each infoset in $S_r$, which represents the difference in value between entering the subgame versus choosing the alternative payoff. Specifically, for each infoset $I_1 \in S_{top}$, the *subgame margin* is

$$M^{\sigma^S}(I_1) = CBV^{\sigma_2}(I_1) - CBV^{\sigma_2^S}(I_1) \tag{2}$$

In Maxmargin solving, a Nash equilibrium $\sigma^S$ for the augmented subgame described in Resolving subgame solving is computed such that the minimum margin over all $I_1 \in S_{top}$ is maximized. Aside from maximizing the minimum margin, the augmented subgames used in Resolving and Maxmargin solving are identical.

Given our base strategy in Coin Toss, Maxmargin solving would result in $P_2$ choosing Heads with probability $\frac{5}{8}$, Tails with probability $\frac{3}{8}$, and Forfeit with probability 0.

The augmented subgame can be solved in a way that maximizes the minimum margin by using a standard LP solver. In order to use iterative algorithms such as the Excessive Gap Technique [23, 11, 18] or Counterfactual Regret Minimization (CFR) [32], one can use the *gadget game* described by Moravcik et al. [21]. Details on the gadget game are provided in the Appendix. Our experiments used CFR.

Maxmargin solving is safe. Furthermore, it guarantees that if every Player 1 best response reaches the subgame with positive probability through some infoset(s) that have positive margin, then exploitability is strictly lower than that of the blueprint strategy. While the theoretical guarantees are stronger, Maxmargin may lead to worse practical performance relative to Resolving when combined with the techniques discussed in Appendix C, due to Maxmargin's greater tendency to overfit to assumptions in the model.

## B  Description of Gadget Game

Solving the augmented subgame described in Maxmargin solving and Reach-Maxmargin solving will not, by itself, necessarily maximize the minimum margin. While LP solvers can easily handle this objective, the process is more difficult for iterative algorithms such as Counterfactual Regret Minimization (CFR) and the Excessive Gap Technique (EGT). For these iterative algorithms, the augmented subgame can be modified into a *gadget game* that, when solved, will provide a Nash equilibrium to the augmented subgame and will also maximize the minimum margin [21]. This gadget game is unnecessary when using distributional alternative payoffs, which is introduced in section C.1.

The gadget game differs from the augmented subgame in two ways. First, all $P_1$ payoffs that are reached from the initial infoset of $I_1 \in S_r$ are shifted by the alternative payoff of $I_1$, and there is longer an alternative payoff. Second, rather than the game starting with a chance node that determines $P_1$'s starting infoset, $P_1$ decides for herself which infoset to begin the game in. Specifically, the game begins with a $P_1$ node where each action in the node corresponds to an infoset $I_1$ in $S_r$. After $P_1$ chooses to enter an infoset $I_1$, chance chooses the precise node $h \in I_1$ in proportion to $\pi_{-1}^{\sigma}(h)$.

By shifting all payoffs in the game by the size of the alternative payoff, the gadget game forces $P_1$ to focus on improving the performance of each infoset over some baseline, which is the goal of Maxmargin and Reach-Maxmargin solving. Moreover, by allowing $P_1$ to choose the infoset in which to enter the game, the gadget game forces $P_2$ to focus on maximizing the minimum margin.

Figure 5 illustrates the gadget game used in Maxmargin and Reach-Maxmargin.

Figure 5: An example of a gadget game in Maxmargin refinement. $P_1$ picks the initial infoset she wishes to enter $S_r$ in. Chance then picks the particular node of the infoset, and play then proceeds identically to the augmented subgame, except all $P_1$ payoffs are shifted by the size of the alternative payoff and the alternative payoff is then removed from the augmented subgame.

## C Modeling Error in a Subgame

In this section we consider the case where we have a good estimate of what the values of subgames would look like in a Nash equilibrium. Unlike previous sections, exploitability might be *higher* than the blueprint when using this method; the solution quality ultimately depends on the accuracy of the estimates used. In practice this approach leads to significantly lower exploitability.

When solving multiple $P_2$ subgames, there is a minimally-exploitable strategy $\sigma_2^*$ that could, in theory, be computed by changing only the strategies in the subgames. ($\sigma_2^*$ may not be a Nash equilibrium because $P_2$'s strategy outside the subgames is fixed, but it is the closest that can be achieved by changing the strategy only in the subgames). However, $\sigma_2^*$ can only be guaranteed to be produced by solving all the subgames together, because the optimal strategy in one subgame depends on the optimal strategy in other subgames.

Still, suppose that we know $CBV^{\sigma_2^*}(I_1)$ for every infoset $I_1 \in S_{top}$ for every subgame $S$. Let $I_{r,1}$ be the infoset in $S_r$ that leads to $I_1$. By setting the $P_1$ alternative payoff for $I_{r,1}$ to $v(I_{r,1}, a_T') = CBV^{\sigma_2^*}(I_1)$, safe subgame solving guarantees a strategy will be produced with exploitability no worse than $\sigma_2^*$. Thus, achieving a strategy equivalent to $\sigma_2^*$ does not require knowledge of $\sigma_2^*$; rather, it only requires knowledge of $CBV^{\sigma_2^*}(I_1)$ for infosets $I_1$ in the top of the subgames.

While we do not know $CBV^{\sigma_2^*}(I_1)$ exactly without knowing $\sigma_2^*$ itself, we may nevertheless be able to produce (or learn) good *estimates* of $CBV^{\sigma_2^*}(I_1)$. For example, in Section 7 we compute the solution to the game of No-Limit Flop Hold'em (NLFH), and find that in perfect play $P_2$ can expect to win about 37 mbb/h[5] (that is, if $P_1$ plays perfectly against the computed $P_2$ strategy, then $P_1$ earns $-37$; if $P_2$ plays perfectly against the computed $P_1$ strategy, then $P_2$ earns 37). An abstraction of the game which is only $0.02\%$ of the size of the full game produces a $P_1$ strategy that can be beaten by 112 mbb/h, and a $P_2$ strategy that can be beaten by 21 mbb/h. Still, the abstract strategy estimates that at equilibrium, $P_2$ can expect to win 35 mbb/h. So even though the abstraction produces a very poor estimate of the *strategy* $\sigma^*$, it produces a good estimate of the *value* of $\sigma^*$. In our experiments, we estimate $CBV^{\sigma_2^*}(I_1)$ by calculating a $P_1$ counterfactual best response *within the abstract game* to $P_2$'s blueprint. We refer to this strategy as $\tilde{CBR}(\sigma_2)$ and its value in an infoset $I_1$ as $\tilde{CBV}^{\sigma_2}(I_1)$.

We then use $C\tilde{B}V^{\sigma_2}(I_1)$ as the alternative payoff of $I_1$ in an augmented subgame. In other words, rather than calculate a $P_1$ counterfactual best response in the full game to $P_2$'s blueprint strategy (which would be $CBR(\sigma_2)$), we instead calculate $P_1$'s counterfactual best response where $P_1$ is constrained by the abstraction.

If the blueprint was produced by conducting $T$ iterations of CFR in an abstract game, then one could instead simply use the final iteration's strategy $\sigma_1^T$, as this converges to a counterfactual best response within the abstract game. This is what we use in our experiments in this paper.

Using estimates of the values of $\sigma^*$ tends to be do better than the theoretically safe options described in Section 4.[6]

## C.1 Distributional Alternative Payoffs

One problem with existing safe subgame-solving techniques is that they may "overfit" to the alternative payoffs, even when we use estimates. Consider for instance a subgame with two different $P_1$ infosets $I_1$ and $I_1'$ at the top. Assume $P_1$'s value for $I_1$ is estimated to be 1, and for $I_1'$ is 10. Now suppose during subgame solving, $P_2$ has a choice between two different strategies. The first sets $P_1$'s value in the subgame for $I_1$ to 0.99 and for $I_1'$ to 9.99. The second slightly increases $P_1$'s value for the subgame for $I_1$ to 1.01 but dramatically lowers the value for $I_1'$ to 0. The safe subgame-solving methods described so far would choose the first strategy, because the second strategy leaves one of the margins negative. However, intuitively, the second strategy is likely the better option, because it is more robust to errors in the model. For example, perhaps we are not confident that 10 is the exact value, but instead believe its true value is normally distributed with 10 as the mean and a standard deviation of 1. In this case, we would prefer the strategy that lowers the value for $I_1'$ to 0.

To address this problem, we introduce a way to incorporate the modeling uncertainty into the game itself. Specifically, we introduce a new augmented subgame that makes subgame solving more robust to errors in the model. This augmented subgame changes the augmented subgame used in subgame Resolving (shown in Figure 3b) so that the alternative payoffs are random variables, and $P_1$ is informed at the start of the augmented subgame of the values drawn from the random variables (but $P_2$ is not). The augmented subgame is otherwise identical. A visualization of this change is shown in Figure 6. As the distributions of the random variables narrow, the augmented subgame converges to the Resolve augmented subgame (but still maximizes the minimum margin when all margins are positive). As the distributions widen, $P_2$ seeks to maximize the sum over all margins, regardless of which are positive or negative.

Figure 6: A visualization of the change in the augmented subgame from Figure 3b when using distributional alternative payoffs.

This modification makes the augmented subgame infinite in size because the random variables may be real-valued and $P_1$ could have a unique strategy for each outcome of the random variable.

Fortunately, the special structure of the game allows us to arrive at a $P_2$ Nash equilibrium strategy for this infinite-sized augmented subgame by solving a much simpler gadget game.

The gadget game is identical to the augmented subgame used in Resolve subgame solving (shown in Figure 3b), except at each initial $P_1$ infoset $I_{r,1} \in S_r$, $P_1$ chooses action $a'_S$ (that is, chooses to enter the subgame rather than take the alternative payoff) with probability $P(X_{I_1} \leq v(I_{r,1}, a'_S))$, where $v(I_{r,1}, a'_S)$ is the expected value of action $a'_S$. (When solving via CFR, it is the expected value on each iteration, as described in CFR-BR [17]). This leads to Theorem 2, which proves that solving this simplified gadget game produces a $P_2$ strategy that is a Nash equilibrium in the infinite-sized augmented subgame illustrated in Figure 6.

**Theorem 2.** *Let $S'$ be a Resolve augmented subgame and $S'_r$ its root. Let $S$ be a Distributional augmented subgame similar to $S'$, except at each infoset $I_{r,1} \in S_r$, $P_1$ observes the outcome of a random variable $X_{I_1}$ and the alternative payoff is equal to that outcome. If CFR is used to solve $S'$ except that the action leading to $S'$ is taken from each $I_{r,1} \in S'_r$ with probability $P(X_{I_1} \leq v^t(I_{r,1}, a'_S))$, where $v^t(I_{r,1}, a'_S)$ is the value on iteration $t$ of action $a'_S$, then the resulting $P_2$ strategy $\sigma_2^{S'}$ in $S'$ is a $P_2$ Nash equilibrium strategy in $S$.*

Another option which also solves the game but has better empirical performance relies on the *softmax* (also known as *Hedge*) algorithm [19]. This gadget game is more complicated, and is described in detail in Appendix D. We use the softmax gadget game in our experiments.

The correct distribution to use for the random variables ultimately depends on the actual unknown errors in the model. In our experiments for this technique, we set $X_{I_1} \sim \mathcal{N}(\mu_{I_1}, s_{I_1}^2)$, where $\mu_I$ is the blueprint value (plus any gifts). $s_{I_1}$ is set as the difference between the blueprint value of $I_1$, and the true (that is, unabstracted) counterfactual best response value of $I_1$. Our experiments show that this heuristic works well, and future research could yield even better options.

# D  Hedge for Distributional Subgame Solving

In this paper we use CFR [32] with Hedge in $S_r$, which allows us to leverage a useful property of the Hedge algorithm [19] to update all the infosets resulting from outcomes of $X_{I_1}$ simultaneously.[7] When using Hedge, action $a'_S$ in infoset $I_{r,1}$ in the augmented subgame is chosen on iteration $t$ with probability $\frac{e^{\eta_t \hat{v}(I_{r,1}, a'_S)}}{e^{\eta_t \hat{v}(I_{r,1}, a'_S)} + e^{\eta_t \hat{v}(I_{r,1}, a'_T)}}$. Where $\hat{v}(I_{r,1}, a'_T)$ is the observed expected value of action $a'_T$, $\hat{v}(I_{r,1}, a'_S)$ is the observed expected value of action $a'_S$, and $\eta_t$ is a tuning parameter. Since, action $a'_S$ leads to identical play by both players for all outcomes of $X$, $\hat{v}(I_{r,1}, a'_S)$ is identical for all outcomes of $X$. Moreover, $\hat{v}(I_{r,1}, a'_T)$ is simply the outcome of $X_{I_1}$. So the probability that $a'_S$ is taken across all infosets on iteration $t$ is

$$\int_{-\infty}^{\infty} \frac{e^{\eta_t \hat{v}(I_{r,1}, a'_S)}}{e^{\eta_t \hat{v}(I_{r,1}, a'_S)} + e^{\eta_t x}} f_{X_{I_1}}(x) dx \tag{3}$$

where $f_{X_{I_1}}(x)$ is the pdf of $X_{I_1}$. In other words, if CFR is used to solve the augmented subgame, then the game being solved is identical to Figure 3b except that action $a'_S$ is always chosen in infoset $I_1$ on iteration $t$ with probability given by (3). In our experiments, we set the Hedge tuning parameter $\eta$ as suggested in [3]: $\eta_t = \frac{\sqrt{\ln(|A(I_1)|)}}{3\sqrt{VAR(I_1)_t}\sqrt{t}}$, where $VAR(I_1)_t$ is the observed variance in the payoffs the infoset has received across all iterations up to $t$. In the subgame that follows $S_r$, we use CFR+ as the solving algorithm.

# E  Full Experimental Results

In tables 3, 4, and 5 we show the full results of our subgame solving experiments on various numbers of buckets.

| Small Flop Hold'em Flop Buckets: | 200 | 2,000 | 30,000 |
|---|---|---|---|
| Blueprint Strategy | 886.9 | 373.7 | 91.28 |
| Unsafe | 146.8 | 39.58 | 5.514 |
| Resolve | 601.6 | 177.9 | 54.07 |
| Maxmargin | 300.5 | 139.9 | 43.43 |
| Reach-Maxmargin | 298.8 | 139.0 | 41.47 |
| Reach-Maxmargin (not split) | 248.7 | 98.07 | 25.88 |
| Estimated | 116.6 | 62.61 | 24.23 |
| Estimated + Distributional | 104.4 | 62.45 | 34.30 |
| Reach-Estimated + Distributional | 102.1 | 57.98 | 22.58 |
| Reach-Estimated + Distributional (not split) | 95.60 | 49.24 | 17.33 |

Table 3: Exploitability (evaluated in the game with no information abstraction) of subgame-solving in small Flop Texas hold'em.

| Large Flop Hold'em Flop Buckets: | 200 | 2,000 | 30,000 |
|---|---|---|---|
| Blueprint Strategy | 283.7 | 165.2 | 41.41 |
| Unsafe | 65.59 | 38.22 | 396.8 |
| Resolve | 179.6 | 101.7 | 23.11 |
| Maxmargin | 134.7 | 77.89 | 19.50 |
| Reach-Maxmargin | 134.0 | 72.22 | 18.80 |
| Reach-Maxmargin (not split) | 130.3 | 66.79 | 16.41 |
| Estimated | 52.62 | 41.93 | 30.09 |
| Estimated + Distributional | 49.56 | 38.98 | 10.54 |
| Reach-Estimated + Distributional | 49.33 | 38.52 | 9.840 |
| Reach-Estimated + Distributional (not split) | 49.13 | 37.22 | 8.777 |

Table 4: Exploitability (evaluated in the game with no information abstraction) of subgame-solving in large Flop Texas hold'em.

| Turn Hold'em Turn Buckets: | 200 | 2,000 | 20,000 |
|---|---|---|---|
| Blueprint Strategy | 684.6 | 465.1 | 345.5 |
| Unsafe | 130.4 | 85.95 | 79.34 |
| Resolve | 454.9 | 321.5 | 251.8 |
| Maxmargin | 427.6 | 299.6 | 234.4 |
| Reach-Maxmargin | 424.4 | 298.3 | 233.5 |
| Reach-Maxmargin (not split) | 333.4 | 229.4 | 175.5 |
| Estimated | 120.6 | 89.43 | 76.44 |
| Estimated + Distributional | 119.4 | 87.83 | 74.35 |
| Reach-Estimated + Distributional | 116.8 | 85.80 | 72.59 |
| Reach-Estimated + Distributional (not split) | 113.3 | 83.24 | 70.68 |

Table 5: Exploitability (evaluated in the game with no information abstraction) of subgame-solving in Turn Texas hold'em.

# F   Scaling of Gifts

To retain the theoretical guarantees of Reach subgame solving, one must ensure that the gifts assigned to reachable subgames do not (in aggregate) exceed the original gift. That is, if $g(I_1)$ is a gift at infoset $I_1$, we must ensure that $CBV^{\sigma_2^*}(I_1) \leq CBV^{\sigma_2}(I_1) + g(I_1)$. In this paper we accomplish this by increasing the margin of an infoset $I_1'$, where $I_1 \sqsubseteq I_1'$, by at most $g(I_1)$. However, empirical performance may improve if the increase to margins due to gifts is scaled up by some factor. In most games we experimented on, exploitability decreased the further the gifts were scaled. However, Figure 7 shows one case in which we observe the exploitability increasing when the gifts are scaled up too far. The graph shows exploitability when the gifts are scaled by various factors. At 0, the algorithm is identical to Maxmargin. at 1, the algorithm is the theoretically correct form of Reach-Maxmargin. Optimal performance in this game occurs when the gifts are scaled by a factor of about 1, 000. Scaling the gifts by 100, 000 leads to performance that is worse than Maxmargin subgame

solving. This empirically demonstrates that while scaling up gifts may lead to better performance in some cases (because an entire gift is unlikely to be used in every subgame that receives one), it may also lead to far worse performance in some cases.

Figure 7: Exploitability in Flop Texas Hold'em of Reach-Maxmargin as we scale up the size of gifts.

## G  Rules for Poker Variants

Our experiments are conducted on heads-up no-limit Texas hold'em (HUNL), as well as smaller-scale variants we call no-limit flop hold'em (NLFH) and no-limit turn hold'em (NLTH). We begin by describing the rules of HUNL.

In the form of HUNL discussed in this paper, each player starts a hand with \$20,000. One player is designated $P_1$, while the other is $P_2$. This assignment alternates between hands. HUNL consists of four rounds of betting. On a round of betting, each player can choose to either fold, call, or raise. If a player folds, that player immediately surrenders the pot to the opponent and the game ends. If a player calls, that players places a number of chips in the pot equal to the opponent's contribution. If a player raises, that player adds more chips to the pot than the opponent's contribution. A round of betting ends after a player calls. Players can continue to go back and forth with raises in a round until one of them runs out of chips.

If either player chooses to raise first in a round, they must raise a minimum of \$100. If a player raises after another player has raised, that raise must be greater than or equal to the last raise. The maximum amount for a bet or raise is the remainder of that player's chip stack, which in our model is \$20,000 at the beginning of a game.

At the start of HUNL, both players receive two private cards from a standard 52-card deck. $P_1$ must place a *big blind* of \$100 in the pot, while $P_2$ must place a *small blind* of \$50 in the pot. There is then a round of betting (the *preflop*), starting with $P_2$. When the round ends, three *community* cards are dealt face up between the players. There is then another round of betting (the *flop*), starting with $P_1$ this time. After the round of betting completes, another community card is dealt face up, and another round of betting commences starting with $P_1$ (the *turn*). Finally, one more community card is dealt face up, and a final betting round occurs (the *river*), again starting with $P_1$. If neither player folds before the final betting round completes, the player with the best five-card poker hand, constructed from their two private cards and the five face-up community cards, wins the pot. In the case of a tie, the pot is split evenly.

NLTH is similar to no-limit Texas hold'em except there are only three rounds of betting (the preflop, flop, and turn) in which there are two options for bet sizes. There are also only four community

cards. NLFH is similar except there are only two rounds of betting (the preflop and flop), and three community cards.

We experiment with two versions of NLFH, one small and one large, which include only a few of the available actions in each infoset. The small game requires 1.1 GB to store the unabstracted strategy as double-precision floats. The large game requires 4 GB. NLTH requires 35 GB to store the unabstracted strategy.

## H   Proof of Theorem 1

*Proof.* Assume $M_r^{\sigma^S}(I_1) \geq 0$ for every infoset $I_1$ and assume $\pi_1^{BR(\sigma_2')}(I_1^*) > 0$ for some $I_1^* \in S_{top}$ and let $\epsilon = M_r(I_1^*)$. Define $\pi_{-1}^{\sigma}(I_1) = \sum_{h \in I_1} \pi_{-1}^{\sigma}(h)$ and define $\pi_{-1}^{\sigma}(I_1, I_1') = \sum_{h \in I_1, h' \in I_1'} \pi_{-1}^{\sigma}(h, h')$.

We show that for every $P_1$ infoset $I_1 \sqsubseteq I_1^*$ where $P(I_1) = P_1$,

$$CBV^{\sigma_2'}(I_1) \leq CBV^{\sigma_2^{-S}}(I_1) +$$
$$\sum_{I_1'' \cdot a'' \sqsubseteq I_1 | P(I_1'') = P_1} \left( \lfloor CBV^{\sigma_2^{-S}}(I_1'') - CBV^{\sigma_2^{-S}}(I_1'', a'') \rfloor \right) - \sum_{h \in I_1, h^* \in I_1^*} \pi_{-1}^{\sigma_2}(h, h^*) \epsilon \quad (4)$$

By the definition of $M_r^{\sigma^S}(I_1^*)$ this holds for $I_1^*$ itself. Moreover, the condition holds for every other $I_1 \in S_{top}$, because by assumption every margin is nonnegative and $\pi_{-1}^{\sigma_2}(I_1, I_1^*) = 0$ for any $I_1 \in S_{top}$ where $I_1 \neq I_1^*$. The condition also clearly holds for any $I_1$ with no descendants in $S$ because then $\pi_{-1}^{\sigma_2}(I_1, I_1^*) = 0$ and $\sigma_2'(h) = \sigma_2^{-S}(h)$ in all $P_2$ nodes following $I_1$. This satisfies the base step. We now move on to the inductive step.

Let $Succ(I_1, a)$ be the set of earliest-reachable $P_1$ infosets following $I_1$ such that $P(I_1') = P_1$ for $I' \in Succ(I_1, a)$. Formally, $I_1' \in Succ(I_1, a)$ if $P(I_1') = P_1$ and $I_1 \cdot a \sqsubseteq I_1'$ and for any other $I_1'' \in Succ(I_1, a)$, $I_1'' \not\sqsubset I_1'$. Then

$$CBV^{\sigma_2'}(I_1, a) = CBV^{\sigma_2^{-S}}(I_1, a) +$$
$$\sum_{I_1' \in Succ(I_1, a)} \pi_{-1}^{\sigma_2'}(I_1, I_1')(CBV^{\sigma_2'}(I_1') - CBV^{\sigma_2^{-S}}(I_1')) \quad (5)$$

Assume that every $I_1' \in Succ(I_1, a)$ satisfies (4). Then

$$CBV^{\sigma_2'}(I_1, a) \leq CBV^{\sigma_2^{-S}}(I_1, a) - \pi_{-1}^{\sigma_2}(I_1, I_1^*)\epsilon +$$
$$\sum_{I_1' \in Succ(I_1, a)} \pi_{-1}^{\sigma_2}(I_1, I_1') \left( \sum_{I_1'' \cdot a'' \sqsubseteq I_1' | P(I_1'') = P_1} \left( \lfloor CBV^{\sigma_2^{-S}}(I_1'') - CBV^{\sigma_2^{-S}}(I_1'', a'') \rfloor \right) \right)$$

$$CBV^{\sigma_2'}(I_1, a) \leq CBV^{\sigma_2^{-S}}(I_1) - \left( CBV^{\sigma_2^{-S}}(I_1) - CBV^{\sigma_2^{-S}}(I_1, a) \right) - \pi_{-1}^{\sigma_2}(I_1, I_1^*)\epsilon +$$
$$\sum_{I_1' \in Succ(I_1, a)} \pi_{-1}^{\sigma_2}(I_1, I_1') \left( \sum_{I_1'' \cdot a'' \sqsubseteq I_1' | P(I_1'') = P_1} \left( \lfloor CBV^{\sigma_2^{-S}}(I_1'') - CBV^{\sigma_2^{-S}}(I_1'', a'') \rfloor \right) \right)$$

Since $\lfloor CBV^{\sigma_2^{-S}}(I_1) - CBV^{\sigma_2^{-S}}(I_1, a) \rfloor \leq CBV^{\sigma_2^{-S}}(I_1) - CBV^{\sigma_2^{-S}}(I_1, a_1)$ so we get

$$CBV^{\sigma_2'}(I_1, a) \leq CBV^{\sigma_2^{-S}}(I_1) - \lfloor (CBV^{\sigma_2^{-S}}(I_1) - CBV^{\sigma_2^{-S}}(I_1, a) \rfloor - \pi_{-1}^{\sigma_2}(I_1, I_1^*)\epsilon +$$
$$\sum_{I_1' \in Succ(I_1, a)} \pi_{-1}^{\sigma_2}(I_1, I_1') \left( \sum_{I_1'' \cdot a'' \sqsubseteq I_1' | P(I_1'') = P_1} \left( \lfloor CBV^{\sigma_2^{-S}}(I_1'') - CBV^{\sigma_2^{-S}}(I_1'', a'') \rfloor \right) \right)$$

$$CBV^{\sigma_2'}(I_1, a) \leq CBV^{\sigma_2^{-S}}(I_1) - \pi_{-1}^{\sigma_2}(I_1, I_1^*)\epsilon +$$
$$\sum_{I_1' \in Succ(I_1, a)} \pi_{-1}^{\sigma_2}(I_1, I_1') \left( \sum_{I_1'' \cdot a'' \sqsubseteq I_1 | P(I_1'') = P_1} \left( \lfloor CBV^{\sigma_2^{-S}}(I_1'') - CBV^{\sigma_2^{-S}}(I_1'', a'') \rfloor \right) \right)$$

$$CBV^{\sigma_2'}(I_1, a_1) \leq CBV^{\sigma_2^{-S}}(I_1) - \pi_{-1}^{\sigma_2}(I_1, I_1^*)\epsilon + \sum_{I_1'' \cdot a'' \sqsubseteq I_1 | P(I_1'') = P_1} \left( \lfloor CBV^{\sigma_2^{-S}}(I_1'') - CBV^{\sigma_2^{-S}}(I_1'', a_1'') \rfloor \right)$$

Since $\pi_1^{BR(\sigma_2')}(I_1^*) > 0$, and action $a$ leads to $I_1^*$, so by definition of a best response, $CBV^{\sigma_2'}(I_1, a) = CBV^{\sigma_2'}(I_1)$. Thus,

$$CBV^{\sigma_2'}(I_1) \leq CBV^{\sigma_2^{-S}}(I_1) - \pi_{-1}^{\sigma_2}(I_1, I_1^*)\epsilon + \sum_{I_1'' \cdot a'' \sqsubseteq I_1 | P(I_1'') = P_1} \left( \lfloor CBV^{\sigma_2^{-S}}(I_1'') - CBV^{\sigma_2^{-S}}(I_1'', a'') \rfloor \right)$$

which satisfies the inductive step.

Applying this reasoning to the root of the entire game, we arrive at $exp(\sigma_2') \leq exp(\sigma_2^{-S}) - \pi_{-1}^{\sigma_2}(I_1^*)\epsilon$. □

# I   Proof of Theorem 2

*Proof.* We prove inductively that using CFR in $S'$ while choosing the action leading to $S'$ from each $I_1 \in S_r'$ with probability $P(X_{I_1} \leq v^t(I_1, a_S'))$ results in play that is identical to CFR in $S$ and CFR-BR [17] in $S_r$, which converges to a Nash equilibrium.

For each $P_2$ infoset $I_2'$ in $S'$ where $P(I_2') = P_2$, there is exactly one corresponding infoset $I_2$ in $S$ that is reached via the same actions, ignoring random variables. Each $P_1$ infoset $I_1'$ in $S'$ where $P(I_1') = P_1$ corresponds to a set of infosets in $S$ that are reached via the same actions, where the elements in the set differ only by the outcome of the random variables. We prove that on each iteration, the instantaneous regret for these corresponding infosets is identical (and therefore the average strategy played in the $P_2$ infosets over all iterations is identical).

At the start of the first iteration of CFR, all regrets are zero. Therefore, the base case is trivially true. Now assume that on iteration $t$, regrets are identical for all corresponding infosets. Then the strategies played on iteration $t$ in $S$ are identical as well.

First, consider an infoset $I_1'$ in $S'$ and a corresponding infoset $I_1$ in $S$. Since the remaining structure of the game is identical beyond $I_1'$ and $I_1$, and because $P_2$'s strategies are identical in all $P_2$ infosets encountered, so the immediate regret for $I_1'$ and $I_1$ is identical as well.

Next, consider a $P_1$ infoset $I_{1,x}$ in $S_r$ in which the random variable $X_{I_1}$ has an observed value of $x$. Let the corresponding $P_1$ infoset in $S_r'$ be $I_1'$. Since CFR-BR is played in this infoset, and since action $a_T'$ leads to a payoff of $x$, so $P_1$ will choose action $a_S'$ with probability 1 if $x \geq a_T'$ and with probability 0 otherwise. Thus, for all infosets in $S_r$ corresponding to $I_1'$, action $a_S'$ is chosen with probability $P(X_{I_1} \leq v(I_1, a_S'))$.

Finally, consider a $P_2$ infoset $I_2$ in $S$ and its corresponding infoset $I_2'$ in $S'$. Since in both cases action $a_T'$ is taken in $S_r$ with probability $P(X_{I_1} \leq v(I_1, a_S'))$, and because $P_1$ plays identically between corresponding infosets in $S$ and $S'$, and because the structure of the game is otherwise identical, so the immediate regret for $I_1'$ and $I_1$ is identical as well. □

## Footnotes

[5]In poker, the performance of one strategy against another depends on how much money is being wagered. For this reason, expected value and exploitability are measured in milli big blinds per hand (mbb/h). A big blind is the amount of money one of the players is required to put into the pot at the beginning of each hand.

[6]It is also possible to combine the safety of past approaches with some of the better performance of using estimates by adding the original Resolve conditions as additional constraints.

[7]Another option is to apply CFR-BR [17] only at the initial $P_1$ nodes when deciding between $a'_T$ and $a'_S$.