[Reviews · NeurIPS 2017]

Reviewer 1



Summary: Considers solving two-player zero-sum games of imperfect information, like poker. Proposes, I think, two new ideas. (1) "Reach" allows the algorithm to make use of any bonus or "gift" that it got at prior steps in the game where the opponent made a mistake. (2) "Nested" proposes to re-solve subgames "on the fly" using more detailed information. That is, the originally solved game may be an abstraction, so when we reach a particular small subgame, maybe we can solve it more exactly in real time. Opinion: I don't know the game-solving and poker-playing type literature very well. The paper seems to me well-written and to advance some interesting and important ideas for that literature. A weakness is relatively little explanation and discussion of the contributions of new techniques except at a technical level, which made it hard for me to evaluate. The "ace in the hole" for the paper is that these techniques recently contributed to the first defeating of top human players in no-limit hold-'em, which is a pretty big landmark for AI in general. Comments: I didn't really get an intuition for why "Reach" helps, i.e. why knowing that P1 made a mistake earlier in the game tree allows us to improve in a certain subgame. I would love to get intuition for this since it seems like a key point. ---- After author response: I found the intuition for "reach" helpful, thanks.

Reviewer 2



This work proposes a new technique "reach subgame solving" to better inform decision making in an imperfect information, two player zero-sum game setting. The main idea is to exploit the knowledge about the difference in payoffs received by the opponent when following paths leading into and away from the information set. The authors also suggest an adaptation that extends the method to multiple independent subgames. The paper is generally well-written, and results have been proved about the low exploitability of the method, or the discrepancy relative to the Nash equilibrium value (in a zero-sum game this value is unique, so the issue of which Nash equilibrium to compare to does not arise). Empirical results have also been provided to substantiate the gains over existing methods - a standard baseline, the unsafe subgame solving method, seems to be performing quite well on all but one benchmarks, but does not come with any theoretical guarantees. The paper also claims to be the first method to have bested four human subjects in heads-up no limit Texas hold'em poker. I wonder if the ideas introduced here could be useful if exploitability were defined not with respect to Nash equilibrium, but some other notions such as (coarse) correlated equilibria? Also, do you think the method could be adapted to non-zero sum (bimatrix games)?

Reviewer 3



The authors present algorithms and experimental results for solving zero-sum imperfect information extensive-form games, with heads-up no-limit Texas Hold'em poker as the motivating application. They begin by reviewing the distinction between "unsafe" and "safe" subgame solving. In unsafe subgame solving, the opponent's strategy outside the subgame, and hence the distribution of the player's state inside the initial information set, is held fixed. In safe subgame solving, the opponent's strategy in the subgame's head is allowed to vary as well, and hence the distribution of true states is responsive to the player's strategy. They then introduce "reach" subgame solving, in which a "gift" (the additional value that the opponent could have obtained by choosing a different subgame) is computed for each path that leads to the subgame. These gifts are added to the margin for each information set. Finally, they introduce nested subgame solving as a method of dealing with large action spaces: the trunk strategy is computed for an abstracted version of the game, but any time the opponent plays an action that is not in the abstraction, a new subgame is generated and solved on-the-fly that contains the action. Experimental results suggest that the two methods combined are extremely effective in practice. Overall I found the exposition well organized and easy to follow, and the performance of the algorithm seems impressive. I have only minor comments about the presentation. 1. Gifts are motivated as "allowing P2 to be less concerned with P1 reaching the subgame along that path" (L.173-174). My guess is that the gift can be added to the margin because it's a measure of how unattractive a branch is to the opponent, and the goal of maxmargin solving is to make the subgame as unattractive as possible to the opponent; but a little more concreteness / hand-holding would be helpful here. 2. At L.297-303, the authors note that in spite of having better theoretical guarantees, the no-split variants seem to always outperform the split variants. They note that this is not necessarily the case if the gifts are scaled up too aggressively; however, that does not seem directly relevant to the point (I don't believe the experimental results include scaling rather than just failure to divide). If the full gift is simply awarded to every subgame, with no division but also no scaling, is exploitability ever worse in practice? If so, can the authors provide an example? If not, do the authors have an intuition for why that might be? At L.210 the authors say that any split is sufficient for the theoretical guarantees, but it seems clear that some splits will leave more on the table than others (as an extreme example, imagine awarding the entire gift to the lowest-probability subgame; this might be close to not using Reach at all). 3. At L.88: "Figure 4a" probably should be "Figure 1".